# Exercise Reduces Glucose Intolerance, Cardiac Inflammation and Adipose Tissue Dysfunction in *Psammomys obesus* Exposed to Short Photoperiod and High Energy Diet

**DOI:** 10.3390/ijms25147756

**Published:** 2024-07-15

**Authors:** Joanne T. M. Tan, Kiara J. Price, Sarah-Rose Fanshaw, Carmel Bilu, Quang Tuan Pham, Anthony Pham, Lauren Sandeman, Victoria A. Nankivell, Emma L. Solly, Noga Kronfeld-Schor, Christina A. Bursill

**Affiliations:** 1Vascular Research Centre, Lifelong Health Theme, South Australian Health and Medical Research Institute, Adelaide, SA 5000, Australia; kiara.price@adelaide.edu.au (K.J.P.); sarahrfanshaw@gmail.com (S.-R.F.); tuanphamquang85@gmail.com (Q.T.P.); anthony.pham@sahmri.com (A.P.); lauren.sandeman@sahmri.com (L.S.); victoria.nankivell@sahmri.com (V.A.N.); emma.solly@sahmri.com (E.L.S.); 2Adelaide Medical School, Faculty of Health and Medical Sciences, University of Adelaide, Adelaide, SA 5005, Australia; 3School of Zoology, Tel Aviv University, Tel Aviv 69978, Israel; carmel.bilu@gmail.com (C.B.); nogaks@tauex.tau.ac.il (N.K.-S.)

**Keywords:** inflammation, cellular hypertrophy, adipocyte differentiation, browning

## Abstract

Circadian disruption causes glucose intolerance, cardiac fibrosis, and adipocyte dysfunction in sand rats (*Psammomys obesus*). Exercise intervention can improve glucose metabolism, insulin sensitivity, adipose tissue function and protect against inflammation. We investigated the influence of exercise on male *P. obesus* exposed to a short photoperiod (5 h light:19 h dark) and high-energy diet. Exercise reduced glucose intolerance. Exercise reduced cardiac expression of inflammatory marker *Ccl2* and *Bax*:*Bcl2* apoptosis ratio. Exercise increased heart:body weight ratio and hypertrophy marker *Myh7*:*Myh6*, yet reduced *Gata4* expression. No phenotypic changes were observed in perivascular fibrosis and myocyte area. Exercise reduced visceral adipose expression of inflammatory transcription factor *Rela*, adipogenesis marker *Ppard* and browning marker *Ppargc1a,* but visceral adipocyte size was unaffected. Conversely, exercise reduced subcutaneous adipocyte size but did not affect any molecular mediators. Exercise increased ZT7 *Bmal1* and *Per2* in the suprachiasmatic nucleus and subcutaneous *Per2*. Our study provides new molecular insights and histological assessments on the effect of exercise on cardiac inflammation, adipose tissue dysfunction and circadian gene expression in *P. obesus* exposed to short photoperiod and high-energy diet. These findings have implications for the protective benefits of exercise for shift workers in order to reduce the risk of diabetes and cardiovascular disease.

## 1. Introduction

Cardiovascular disease (CVD) and type 2 diabetes mellitus (T2DM) are increasing global epidemics. CVD is the leading cause of death worldwide, and it is a frequent comorbidity of T2DM, with almost one in three people with T2DM also suffering from CVD [1]. Furthermore, people with diabetes or considered pre-diabetic are at higher risk of developing CVD than those without [2,3]. T2DM causes abnormalities in adipose tissue (e.g., insulin-resistant adipocytes, adipocyte hypertrophy, impaired metabolism/differentiation) and cardiac tissue (e.g., inflammation, myocardial fibrosis, myocyte hypertrophy) [4,5]. These alterations increase the risk of cardiometabolic disorders and heart failure.

Circadian rhythms are biological processes that occur within 24-h periods. They are directly or indirectly controlled by circadian clocks, which are regulated by a host of transcriptional factors that interact with each other via negative feedback loops [6]. The circadian system is responsible for metabolic homeostasis and modulates and synchronises physiological and metabolic processes, such as sleep/wake and fasting/feeding cycles [7]. The circadian system consists of central and peripheral molecular clocks in all nucleated cells [8]. The central circadian clock, the suprachiasmatic nucleus (SCN) in the hypothalamus, is influenced predominantly by light but also by feeding and exercise to maintain daily rhythms and genetically regulate peripheral clocks in almost every other cell in the body [7]. Disruption of circadian rhythms has wide-ranging pathophysiological consequences, including impaired glucose regulation, increased obesity risk, metabolic syndrome, inflammation, CVD and T2DM [9].

Exercise intervention has been reported to improve glucose metabolism, insulin sensitivity, adipose tissue function and protect against inflammation [10,11]. Consequently, exercise is found to effectively reduce the risk of T2DM and CVD [12]. Furthermore, exercise has been deemed an external synchroniser of the circadian rhythm [9]. As an external synchroniser, its impact occurs in the peripheral clocks, which are in nearly every cell, rather than in the suprachiasmatic nuclei (SCN) of the brain [7]. The molecular clock cells in the SCN use electrical and chemical signals to synchronise their activity. However, the impact of long-lasting regular exercise on SCN clock cell coordination and communication remains unresolved [13].

The *Psammomys obesus* sand rat is a unique diurnal polygenic rodent model that spontaneously develops T2DM when fed a high-energy diet. Recently, we have shown that circadian disruption impairs glucose tolerance, increases cardiac fibrosis and drives adipocyte dysfunction in this species [4,5,14]. In the current study, we sought to determine the effect of exercise on male *P. obesus* exposed to a short photoperiod (5 h light:19 h dark) and a high-energy diet. The Exercise group had voluntary access to a running wheel, which allowed them to perform low-moderate-intensity exercise for 10 weeks.

## 2. Results

### 2.1. Exercise Prevented Circadian Disruption-Induced Glucose Intolerance

Exercise did not affect body weights and fasting baseline glucose levels (Table 1). Each animal underwent an oral glucose tolerance test (OGTT) at ZT2 to measure changes in the circulatory clearance of glucose after 9 weeks of exercise. Circadian disruption caused glucose intolerance in the control group, as witnessed by the significant increase in blood glucose levels 120 min post-bolus glucose administration (139%, *p* < 0.001, Figure 1). In contrast, for the *P. obesus* who had the exercise running wheel, the effects of short photoperiod exposure on glucose tolerance were mitigated, and there was no increase in glucose levels after 120 min. Glucose levels were significantly lower in the exercise group than the control group at the 120-min time point (33%, *p* < 0.001, Figure 1).

### 2.2. Exercise Reduced Inflammation and Apoptosis in Cardiac Tissue but Had No Effect on Myocardial Perivascular Fibrosis

Hyperglycemia-induced inflammation drives the development of myocardial perivascular fibrosis [15]. *P. obesus* in the exercise group had significantly lower mRNA levels of inflammatory chemokine *Ccl2* (40.8 ± 11.4%, Figure 2a) compared to the control group (100.0 ± 43.3%, *p* < 0.001). Exercise also caused a non-significant trend for a reduction in *Tgfb1* mRNA (Exercise: 72.3 ± 20.1% vs. Control: 100.0 ± 43.2%, *p* = 0.0874, Figure 2b). To determine whether exercise affected apoptosis in cardiac tissues, the mRNA levels of pro-apoptotic gene *Bax* and anti-apoptotic gene *Bcl2* were measured. There was a slight but significant decrease in the *Bax*:*Bcl2* apoptosis ratio in the exercise group compared to controls (Exercise: 96.4 ± 0.7% vs. Control: 100.0 ± 4.5%, *p* < 0.05, Figure 2c), demonstrating that exercise decreases apoptosis in cardiac tissue. However, no change in perivascular collagen deposition, a marker of fibrosis, was observed around the myocardial vessels between the exercise and control groups (Figure 2d).

### 2.3. Exercise Increased the Heart:Body Weight Ratio and Hypertrophy Marker Myh7:Myh6, Yet Reduced Transcription Factor Gata4

It is well-established that exercise increases heart weight and causes cardiac tissue hypertrophy. The heart:body weight ratio was significantly higher in *P. obesus* in the exercise group than in the control (Exercise: 133.3 ± 16.9% vs. Control: 100.0 ± 14.7%, *p* < 0.001, Figure 3a). Diabetes is known to be linked to cardiac hypertrophy in *P. obesus* [16]. Cardiac hypertrophy was assessed by measuring the ratio of hypertrophy genes *Myh7*: *Myh6*. Consistent with the increased heart:body weight ratio, there was a slight but significant increase in the *Myh7*:*Myh6* mRNA ratio (Exercise: 106.6 ± 2.5% vs. Control: 100.0 ± 6.3%, *p* < 0.01, Figure 3b). Despite this, exercise induced a significant reduction in *Gata4*, the transcription factor that controls these hypertrophy markers (Exercise: 41.3 ± 20.1% vs. Control: 100.0 ± 40.9%, *p* < 0.001, Figure 3c), and there were no differences in cardiomyocyte size (Figure 3d).

### 2.4. Exercise Reduced Visceral Adipose Expression of the Inflammatory Transcription Factor Rela

Inflammation is an underlying cause of adipose dysfunction, and hyperglycaemia contributes to a pro-inflammatory environment [17]. We recently showed that circadian disruption combined with a high-energy diet stimulates a pro-inflammatory environment in visceral adipose tissue [5]. In this study, exercise significantly decreased gene expression of *Rela*, the active p65 subunit of NFκB (Exercise: 34.5 ± 22.5% vs. Control: 100.0 ± 58.1%, *p* < 0.01, Figure 4a). However, no differences were observed in the subcutaneous adipose tissue (Figure 4b).

### 2.5. Exercise Reduced Visceral Adipose Expression of Adipogenesis Markers

Increased adipocyte differentiation is associated with improved adipocyte function. We next explored the effect of exercise on the gene expression of *Ppard*, *Pparg* and *Cebpa*, critical transcription factors involved in adipogenesis [18]. Exercise significantly reduced visceral *Ppard* expression (Exercise: 27.2 ± 13.7% vs. Control: 100.0 ± 94.5%, *p* < 0.01, Figure 5a). Visceral expression of *Pparg* (Figure 5b) and *Cebpa* (Figure 5c) were approximately 50% lower in the exercise group, although these did not quite reach statistical significance (*p* = 0.0674 and *p* = 0.1058 respectively). In contrast, no differences were observed in the subcutaneous adipose tissue (Figure 5d–f).

### 2.6. Exercise Reduced Visceral Adipose Expression of Browning Markers

Promoters of adipocyte browning trigger the transdifferentiation of white adipocytes into brown/beige adipocytes, forming a more thermogenic fat phenotype that oxidises glucose and lipids via UCP1-mediated thermogenesis [19]. Browning is known to improve insulin sensitivity and protect against metabolic dysfunction [20]. In this study, exercise significantly reduced visceral expression of browning marker *Ppargc1a* (Exercise: 12.8 ± 7.0% vs. Control: 100.0 ± 120.9%, *p* < 0.01, Figure 6a) with a similar non-significant trend seen in *Ucp1* levels (Figure 6b). In contrast, non-significant increases were observed in the subcutaneous adipose tissue of the exercise group (Figure 6c,d).

### 2.7. Exercise Reduced Subcutaneous Adipocyte Area

Adipocyte hypertrophy is an established indicator of the degree of metabolic impairment/disturbance in patients [21]. Increased adipocyte size is associated with key characteristic parameters that underpin T2DM, including insulin resistance and a pro-inflammatory phenotype [22]. While we observed no differences in visceral adipocyte area (Figure 7a), exercise significantly reduced subcutaneous adipocyte area (Exercise: 1931 ± 544 µm^2^ vs. Control: 3180 ± 1292 µm^2^, *p* < 0.05, Figure 7b).

### 2.8. Exercise Increased Circadian Genes at ZT7

We next explored the effects of exercise on the expression of clock genes *Bmal1*, *Clock* and *Per2*. In the suprachiasmatic nucleus (SCN), exercise significantly increased ZT7 *Bmal1* mRNA (Exercise: 137.7 ± 51.3% vs. Control: 100.0 ± 16.5%, *p* < 0.05, Figure 8a). SCN expression of *Clock* expression was approximately 20% higher in the exercise group, although this did not quite reach statistical significance (*p* = 0.1006, Figure 8b). Exercise significantly increased ZT7 *Per2* mRNA in the suprachiasmatic nucleus (Exercise: 143.8 ± 46.7% vs. Control: 100.0 ± 25.3%, *p* < 0.05, Figure 8c). In the prefrontal cortex (PFC), no differences were observed in gene expression of *Bmal1* (Figure 8d) and *Clock* (Figure 8e). We observed non-significant 40% increases in *Per2* expression in both the prefrontal cortex (*p* = 0.0994, Figure 8f) and heart (*p* = 0.0532, Figure 8g). While no differences were seen in visceral *Per2* expression (Figure 8h), subcutaneous *Per2* was 2-fold higher in the exercise group (Exercise: 201.6 ± 75.3% vs. Control: 100.0 ± 45.2%, *p* < 0.01, Figure 8i).

## 3. Discussion

Glucose intolerance caused by circadian disruption predisposes individuals to the development of T2DM, which promotes myocardial inflammation, fibrosis, and adipocyte dysfunction. Using the unique animal model of *P. obesus,* the effects of exercise were tested in animals exposed to a short photoperiod (i.e., circadian disruption) and a high-energy diet. We report the following important findings: exercise (1) protected against circadian disruption-induced glucose intolerance, (2) reduced cardiac inflammation and apoptosis, and (3) increased cardiac hypertrophy markers and heart weight. (4) In visceral adipose tissue, exercise reduced inflammation, adipogenesis and browning markers but did not affect adipocyte area. (5) Exercise reduced subcutaneous adipocyte area but did not affect gene expression. (6) Finally, at ZT7, exercise increases *Per2* expression in the SCN and subcutaneous adipose tissue, with trends for an increase in the PFC and heart but not visceral adipose tissue.

*P. obesus* has been reported to exhibit impaired glucose tolerance in response to circadian disruption, which is further exacerbated by the consumption of a high-energy diet [4,14]. Circadian disruption causes decreased insulin sensitivity and impaired glucose homeostasis, both of which are associated with early stages of T2DM [15]. Exercise intervention has been shown to improve these compromised metabolic processes in *P. obesus* [23]. Consistent with this, the current study found that exercise protected against glucose intolerance. Unfortunately, we were unable to measure any other indices of glucose tolerance in these animals, such as adipose tissue accumulation and HbA1c levels. Whilst the mechanisms for this remain unclear, as a non-photic cue for the peripheral clocks of the circadian system, exercise may be able to restore rhythm [23,24]. Alternatively, exercise also induces positive changes in the adipose tissue, which is a major endocrine organ [1] and may, therefore, contribute to correcting circadian disruption-induced glucose homeostasis.

Inflammation driven by hyperglycemia underlies the development of myocardial perivascular fibrosis, which is associated with heart failure [15]. Despite the reduction seen in inflammatory chemokine *Ccl2*, there was no significant change in perivascular collagen deposition around the myocardial vessels with exercise. We previously found that circadian disruption significantly reduced cardiac inflammatory and fibrotic gene expression in *P. obesus* [4]. A reduction in perivascular fibrosis was also noted; however, this was only at the later timepoint of a 20-week intervention and did not occur in the 8-week cohort. This indicates that myocardial perivascular fibrosis may take a longer period of time to develop. Therefore, as the duration of our study was 10 weeks, we postulate that the changes in inflammatory gene expression seen in the exercise group have not yet induced changes in the amount of perivascular fibrosis in the myocardial tissue. This is supported by the observed non-significant trend for a reduction in *Tgfb1*, a gene that has a role in promoting fibrosis.

The *Bax*:*Bcl2* ratio is an indicator of apoptosis that can be triggered under diabetic conditions in cardiac tissue [25]. We found that *Bax*:*Bcl2* was significantly reduced in cardiac tissue following exercise. This is indicative of a protective mechanism caused by exercise that prevents cardiomyocyte apoptosis induced by the high-energy diet and circadian disruption [25]. Consistent with this, previous studies have reported that exercise improves cardiac function and attenuates myocardial inflammation and apoptosis in mice [26,27].

Aerobic exercise induces physiological myocardial hypertrophy, with a thickening of the myocardium and increased myofilament attachment points [28]. Our study found that exercise increased the heart:body weight ratio concomitant with an increase in hypertrophy marker *Myh7*:*Myh6* ratio. Despite this, no change was observed in cardiomyocyte size between the exercise and control groups. This apparent anomaly between heart weight and cardiomyocyte size suggests that the cardiomyocytes have undergone increased proliferation in response to exercise, although cardiomyocyte renewal is reported to be relatively limited [29]. Our previous study of exercise intervention in *P. obesus* observed that heart weight increased with exercise via thickening of the left ventricle wall, suggesting that this morphological change to the cardiac tissue was a physiological adaptation to exercise rather than a pathological process driven by hyperglycemia [23]. Thickening of the left ventricular wall was not assessed in the current study, but it is possible that this is the underlying cause of increased heart weight in the exercise cohort. We have previously shown that *P. obesus* exposed to a short photoperiod and/or fed a high-energy diet for 8 weeks exhibited a significant increase in hypertrophy gene expression without concomitant cardiomyocyte hypertrophy [4]. This aligns with our study and suggests that phenotypic changes to the cardiac tissue occur after altered gene expression. Interestingly, despite the upregulation of hypertrophy genes, we observed a decrease in gene expression of *Gata4*, the transcription factor for *Myh7* and *Myh6*. Similar results were observed in a study investigating diabetic mice, though the mechanism for this was unclear [30]. Another study showed that hypertrophic mouse hearts had a significant increase in GATA4 protein levels without a corresponding increase in *Gata4* mRNA levels [31].

We previously found that circadian disruption and hyperglycemia increased adipose inflammation, a key characteristic that drives adipocyte dysfunction and obesity [5]. Exercise improves glucose tolerance and insulin insensitivity, which is a protective mechanism against inflammation [10]. Thus, *Rela* decrease may be indicative of a protective mechanism against the deleterious effects of circadian disruption and visceral adipose inflammation, a finding that has previously been reported using this model [23]. Consistent with this, we found that exercise reduced visceral *Rela* expression. Given the decrease in adipose inflammation, it is likely that whole-body inflammation may also be reduced. Unfortunately, we were unable to measure circulating inflammatory mediators due to the limited availability of species-specific ELISAs.

We found that exercise only decreased visceral expression of adipogenesis marker *Ppard*, with no changes in *Pparg* or *Cebpa*, the key transcription factors that drive adipocyte differentiation. PPARδ primarily regulates lipid metabolism but has also been shown to promote adipogenesis. Activation of PPARδ shifts energy production from glycolysis to fatty acid oxidation, which reduces circulating lipids and prevents adipocyte hypertrophy and lipid accumulation [32]. Transdifferentiation of white adipocytes into the more metabolically active brown/beige adipocytes is a protective mechanism against adipocyte insulin resistance and dysfunction [20]. Unexpectedly, we observed a decrease in browning marker *Ppargc1a* with a similar trend in *Ucp1*. While we have previously shown that *Ppargc1a* expression also declined in *P. obesus* fed a high energy diet [5], this contradicts another study that showed increased visceral expression of browning markers with exercise [11]. Furthermore, *Ppargc1a* is a co-activator of *Pparg,* which enhances the transdifferentiation of adipocytes from white to beige/brown and brown fat thermogenesis [33,34]. Therefore, without first influencing the first step in the pathway, none of the cascading effects of *Ppargc1a* were rectified, leading to a decrease in browning marker expression [34]. Despite these significant changes at a molecular level, exercise did not affect the visceral adipocyte area. This could again be attributed to the short exercise duration, and perhaps a longer duration may yield phenotypic changes. Interestingly, exercise reduced subcutaneous adipocyte area despite a lack of change in molecular mediators. Numerous studies have shown that exercise reduces both visceral and subcutaneous adipocyte size [35,36,37,38]. Given the promising molecular changes seen in the visceral fat, it is possible that a longer duration may drive phenotypic effects.

Circadian clocks are self-sustained autoregulatory transcriptional/translation feedback loop cycles with a free-running period of approximately 24 h [39,40,41]. The circadian clock mechanism is driven by two transcription factors, CLOCK and BMAL1 [41,42,43]. Upon heterodimerisation, the CLOCK/BMAL1 heterodimer binds to E-boxes in target genes, leading to the induction of its repressors period (PER1/2/3), cryptochrome (CRY1/2), and REV-ERB (REV-ERBα/β) or activators retinoic acid receptor-related orphan receptors (RORα/β) [44]. Impaired rhythms in these circadian clock genes will affect the expression of multiple clock-controlled genes, such as metabolic genes, signalling genes, and epigenetic regulators [44,45,46,47]. We found significant increases in *Bmal1* and *Per2* expression in the SCN at ZT7. SCN *Clock* expression was also 20% higher in the exercise group, although this did not quite reach statistical significance (*p* = 0.1006). These findings suggest that the greatest impact of exercise is on the SCN. We also observed a significant increase in subcutaneous *Per2*, which might contribute to the changes in adipocyte size seen. Additionally, there was also a non-significant increase in *Per2* levels in both the PFC (*p* = 0.0994) and cardiac tissue (*p* = 0.0532). These findings may be indicative of a move towards improved exercise-induced rhythmicity, particularly in the cardiac and subcutaneous adipose tissue. In contrast, there was no change in *Per2* in the visceral adipose tissue, perhaps explaining the divergence in effects of exercise on the cardiac tissue versus visceral adipose tissue. This may indicate that in the cardiac tissue, exercise had an effect on rhythmicity that had been impaired due to circadian disruption.

## 4. Materials and Methods

### 4.1. Animal Studies

All experimental procedures followed the NIH guidelines for the care and use of laboratory animals and were approved by the Institutional Animal Care and Use Committee (IACUC) of Tel Aviv University (Permit Number: L15055, 2017). This study was performed on HsdHu diabetes-prone male sand rats (*Psammomys obesus*, 6–7 months old) (Appendix A). Animals were initially maintained on a low-energy diet (Product No.: 1078, Koffolk Ltd., Tel Aviv, Israel) and neutral photoperiod (12 h light:12 h dark) to prevent the development of diabetes [23]. The normal day/night cycle for *P. obesus* is 12 h light:12 h dark. After 3 weeks of acclimation, the animals were assigned to experimental groups based on their weights and blood glucose levels to avoid baseline bias. Adult male sand rats (n = 24) were exposed to a short photoperiod (5 h light:19 h dark, i.e., circadian disruption) and a high-energy diet ad libitum (standard rodent food; 21% protein, 4% fat, 4% crude fibre; Product No.: 2018; Koffolk Ltd.) which contains a higher caloric density and is known to drive T2DM development in *P. obesus* [4,5,14,23]. The experimental exercise group (n = 12) had voluntary access to a running wheel in their individual cages for 10 weeks; the animals in the control group (n = 12) did not have access to a running wheel in their individual cages. Generally, the animals performed low-moderate-intensity exercise that would be aerobic (not anaerobic) and/or endurance and not resistance exercise.

At week 9, animals were subjected to an oral glucose tolerance test at ZT2. Animals received a bolus of glucose at 2 mg/kg body weight via gastric gavage. Blood was collected via tail vein nick at 0 min and 120 min post glucose bolus administration to assess blood glucose levels using a glucometer [4,5,14,23]. At week 10, the sand rats were euthanised around ZT7 (during the dark phase), and blood was collected. The heart, visceral and subcutaneous adipose tissues were collected. Additionally, the brain was dissected to collect the suprachiasmatic nucleus (SCN) and prefrontal cortex (PFC).

### 4.2. Gene Expression Analysis

Total RNA was extracted from all tissues using TRI^®^ reagent (Sigma-Aldrich, St. Louis, MO, USA). RNA concentration and quality were assessed spectrophotometrically. 500 ng of total RNA was converted to cDNA using the iScript cDNA Synthesis Kit (Bio-Rad, Hercules, CA, USA). Quantitative real-time PCR was performed to assess (1) cardiac expression of *Ccl2*, *Tgfb1*, *Bax*, *Bcl2*, *Myh6*, *Myh7*, *Gata4*, *Per2* and *Cyclophilin*; (2) adipose expression of *Rela*, *Ppargc1a*, *Ucp1*, *Ppard*, *Pparg*, *Cebpa*, *Per2* and *Cyclophilin*; and (3) SCN and PFC expression of *Per2* and *Cyclophilin* using previously published primers [4,5,48,49,50]. Relative gene expression was calculated using the ^ΔΔ^Ct method, normalised to *Cyclophilin* and control group.

### 4.3. Histological Analysis

The right ventricle of the hearts was formalin-fixed and paraffin-embedded, then sectioned on a microtome at 5 µm. Two sections were taken ~50 µm apart and stained with Masson’s Trichrome (Sigma, St. Louis, MO, USA). Under 400× total magnification, individual vessels were photographed. Five photos per section were selected in two sections. Perivascular cardiac fibrosis was determined by calculating a ratio of the area of collagen surrounding the vessels divided by the vessel area [4,48]. Two sections were also taken ~50 µm apart and stained with Hematoxylin and Eosin (H&E, Thermofisher, Waltham, MA, USA) and then imaged to define cell morphology. Cardiomyocytes with centralised nuclei were focal regions for analysis. Cardiomyocyte size was determined by calculating the area of the eosin stain (cytoplasm) by the number of hematoxylin-stained nuclei [4]. All images were taken using an AxioLab microscope attached to a camera (Zeiss, Oberkochen, Baden-Württemberg, Germany) and then analysed with Image-Pro Premier 9.2.

A portion of visceral and subcutaneous adipose tissue was formalin-fixed, paraffin-embedded, and sectioned using a microtome at 5 µm thickness. Two sections were taken ~50 µm apart and stained with H&E to define adipocyte morphology [5]. Two fields of view were imaged per section (400× magnification) using an AxioLab microscope attached to a camera (Zeiss). The area of the adipocytes was determined by manual tracing of the interior of the adipocyte, and results were analysed as mean area per animal using ZEN lite 2.3 [5].

### 4.4. Statistics

Data is presented as mean ± SD. The normal distribution of data was determined using the Shapiro–Wilk test. Differences between groups were calculated using either an unpaired *t*-test or the Mann–Whitney test. Significance was set at a two-sided *p* < 0.05.

## 5. Conclusions

Our study provides a first-time assessment of the effect of exercise on cardiac inflammation, adipose tissue dysfunction and circadian gene expression in *P. obesus* exposed to short photoperiod and high energy diet. Collectively, we show that exercise reduces glucose intolerance, cardiac inflammation, and adipose tissue dysfunction in *P. obesus* when exposed to a short photoperiod. Evidently, exercise imparted a greater beneficial influence on the cardiac than visceral adipose tissue. This observation is supported by the divergence in circadian clock gene responses. These findings have clinical implications for shift workers experiencing circadian disruption, suggesting that exercise is an effective intervention to prevent T2DM and cardiovascular disease.

## Figures and Tables

**Figure 1 ijms-25-07756-f001:**
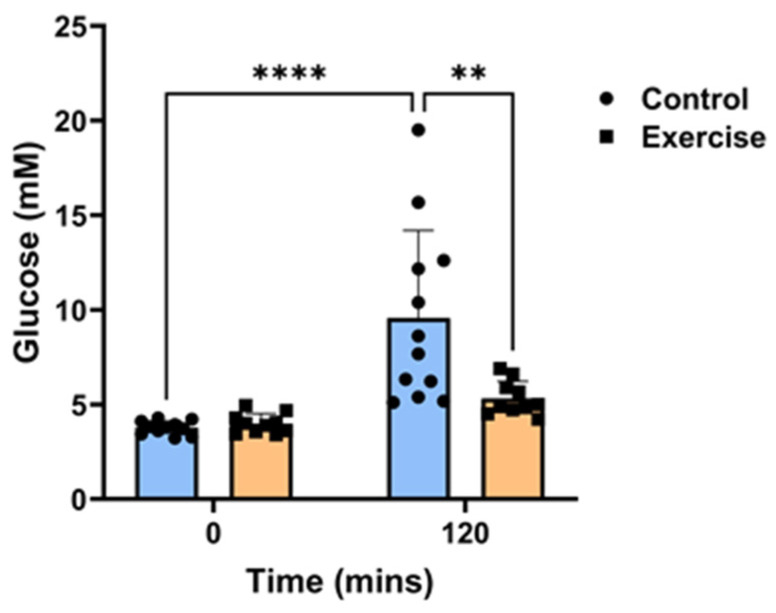
Exercise prevented circadian disruption-induced glucose intolerance. Male *P. obesus* (n = 12/group) were split into two groups: (1) control (no running wheel) and (2) exercise (running wheel) and exposed to a short photoperiod (5 h light:19 h dark) and fed a high energy diet for 10 weeks. Oral glucose tolerance tests were performed in week 9 at ZT2, where animals received a bolus administration of glucose (2 g/kg body weight). Blood glucose levels were measured 120 min post-glucose administration. Data presented as mean ± SD. ** *p* < 0.01, **** *p* < 0.0001 by two-way ANOVA (Šídák’s *post hoc* comparison).

**Figure 2 ijms-25-07756-f002:**
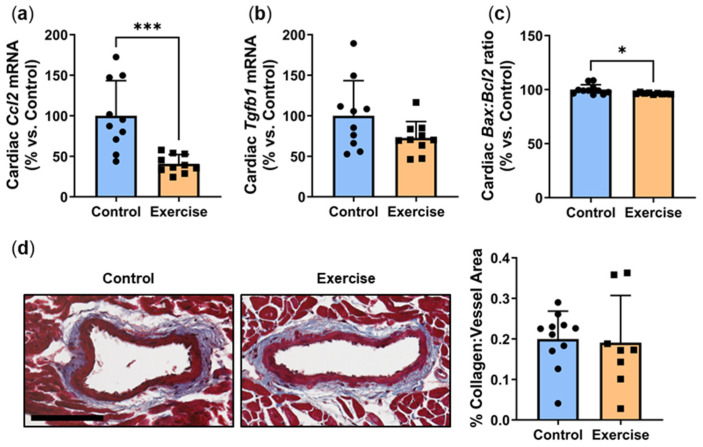
Exercise reduced inflammation and apoptosis in cardiac tissue but had no effect on myocardial perivascular fibrosis. Male *P. obesus* (n = 12/group) were split into two groups: (1) control (no running wheel) and (2) exercise (running wheel) and exposed to a short photoperiod (5 h light:19 h dark) and fed a high energy diet for 10 weeks. Cardiac expression of (**a**) *Ccl2*, (**b**) *Tgfb1* and (**c**) *Bax*:*Bcl2* ratio, normalised using the ^ΔΔ^Ct method to *Cyclophilin* and controls. (**d**) Representative images of Masson’s trichrome-stained hearts depicting collagen (blue) deposition surrounding the vessels. Perivascular fibrosis was analysed by determining the area of blue staining around selected vessels and normalised to the vessel area. Cropped images are used in the figure, while the original, uncropped images are presented in Appendix A. Data presented as mean ± SD. * *p* < 0.05, *** *p* < 0.001 by unpaired *t*-test or Mann–Whitney test.

**Figure 3 ijms-25-07756-f003:**
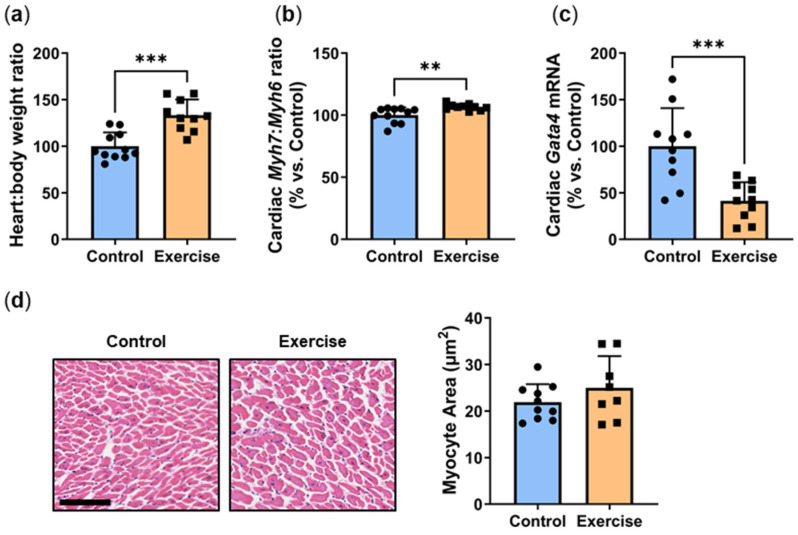
Exercise increased the heart:body weight ratio and hypertrophy marker *Myh7*:*Myh6*, yet reduced transcription factor *Gata4*. Male *P. obesus* (n = 12/group) were split into two groups: (1) control (no running wheel) and (2) exercise (running wheel) and exposed to a short photoperiod (5 h light:19 h dark) and fed a high energy diet for 10 weeks. (**a**) Heart:body weight ratio (**b**) Cardiac *Myh7*:*Myh6* ratio was determined by qPCR. (**c**) Cardiac *Gata4* mRNA, normalised using the ^ΔΔ^Ct method to *Cyclophilin* and controls. (**d**) Representative images of H&E-stained hearts. Myocyte area was analysed by determining the area of eosin staining divided by the number of nuclei/images. Cropped images are used in the figure, while the original, uncropped images are presented in Appendix A. Data presented as mean ± SD. ** *p* < 0.01, *** *p* < 0.001 by unpaired *t*-test or Mann–Whitney test.

**Figure 4 ijms-25-07756-f004:**
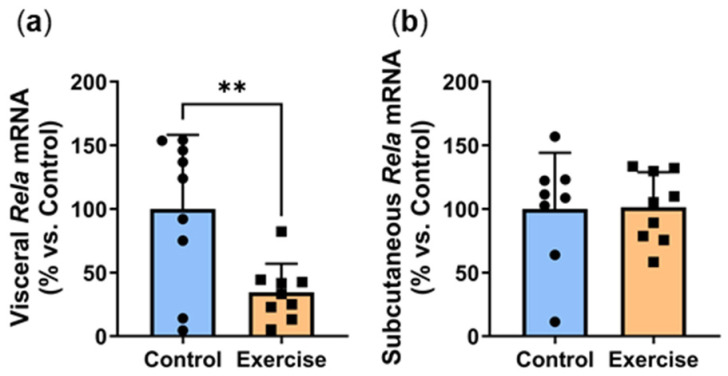
Exercise reduced visceral adipose expression of the inflammatory transcription factor *Rela*. Male *P. obesus* (n = 12/group) were split into two groups: (1) control (no running wheel) and (2) exercise (running wheel) and exposed to a short photoperiod (5 h light:19 h dark) and fed a high energy diet for 10 weeks. (**a**) Visceral and (**b**) subcutaneous *Rela* mRNA, normalised using the ^ΔΔ^Ct method to *Cyclophilin* and controls. Data presented as mean ± SD. ** *p* < 0.01 by unpaired *t*-test.

**Figure 5 ijms-25-07756-f005:**
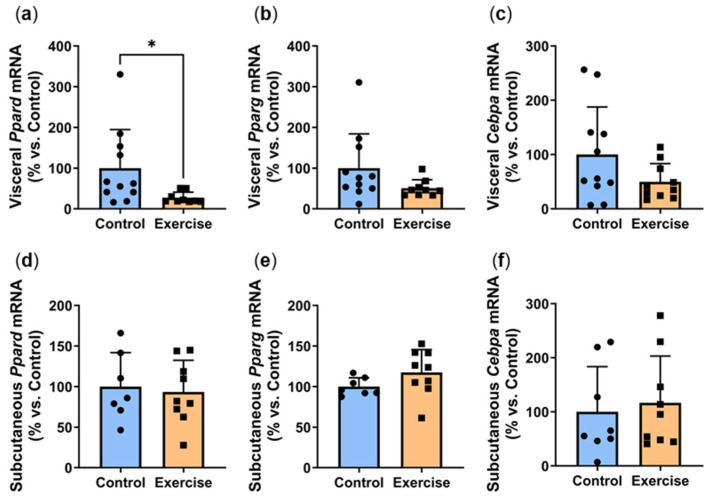
Exercise reduced visceral adipose expression of adipogenesis markers. Male *P. obesus* (n = 12/group) were split into two groups: (1) control (no running wheel) and (2) exercise (running wheel) and exposed to a short photoperiod (5 h light:19 h dark) and fed a high energy diet for 10 weeks. (**a**) Visceral *Ppard*, (**b**) visceral *Pparg*, (**c**) visceral *Cebpa*, (**d**) subcutaneous *Ppard*, (**e**) subcutaneous *Pparg*, (**f**) subcutaneous *Cebpa* mRNA, normalised using the ^ΔΔ^Ct method to *Cyclophilin* and controls. Data presented as mean ± SD. * *p* < 0.05 by Mann–Whitney test. Statistical analyses were performed using either the unpaired *t*-test or the Mann–Whitney test.

**Figure 6 ijms-25-07756-f006:**
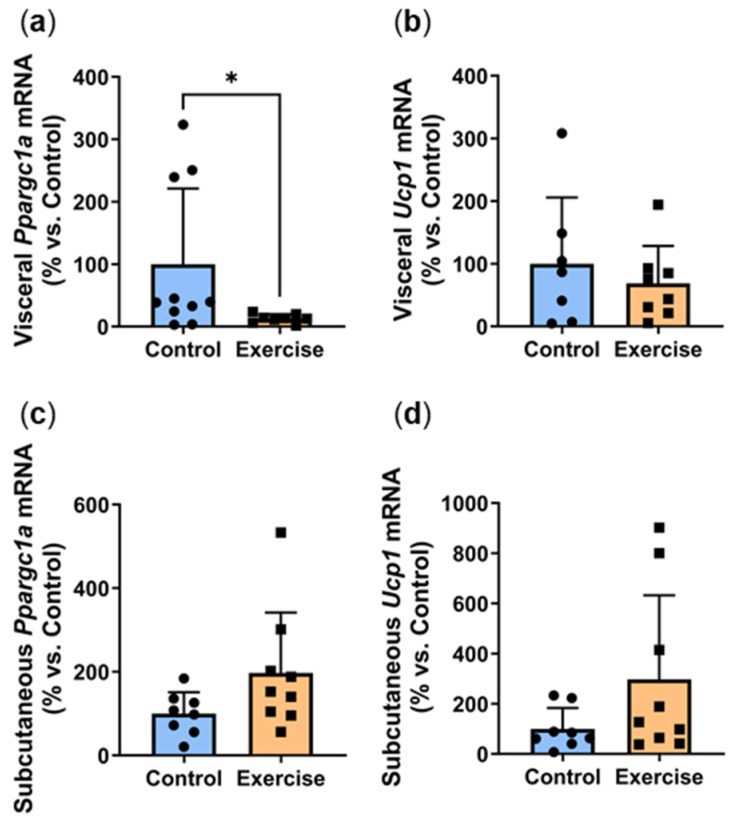
Exercise reduced visceral adipose expression of browning markers. Male *P. obesus* (n = 12/group) were split into two groups: (1) control (no running wheel) and (2) exercise (running wheel) and exposed to a short photoperiod (5 h light:19 h dark) and fed a high energy diet for 10 weeks. (**a**) Visceral *Ppargc1a*, (**b**) visceral *Ucp1*, (**c**) subcutaneous *Ppargc1a*, (**d**) subcutaneous *Ucp1* mRNA, normalised using the ^ΔΔ^Ct method to *Cyclophilin* and controls. Data presented as mean ± SD. * *p* < 0.05 by Mann–Whitney test. Statistical analyses were performed using either the unpaired *t*-test or the Mann–Whitney test.

**Figure 7 ijms-25-07756-f007:**
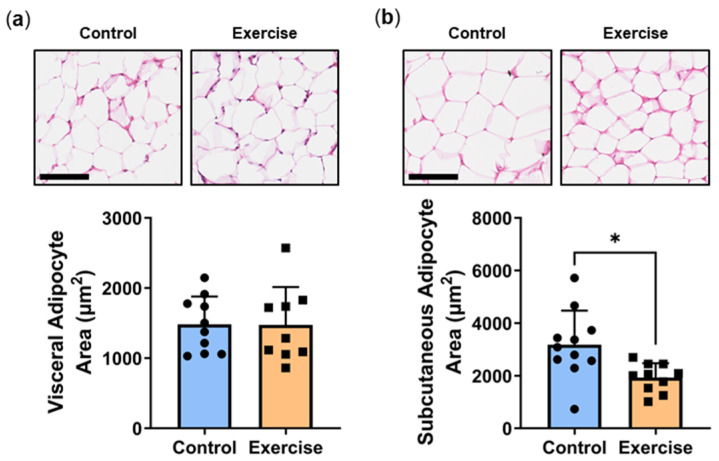
Exercise reduced subcutaneous adipocyte area. Male *P. obesus* (n = 12/group) were split into two groups: (1) control (no running wheel) and (2) exercise (running wheel) and exposed to a short photoperiod (5 h light:19 h dark) and fed a high energy diet for 10 weeks. Representative images and corresponding average (**a**) visceral and (**b**) subcutaneous cell area was calculated from the average cell area measurement/animal per group. Cropped images are used in the figure, while the original, uncropped images are presented in Appendix A. Data presented as mean ± SD. * *p* < 0.05 by unpaired *t*-test.

**Figure 8 ijms-25-07756-f008:**
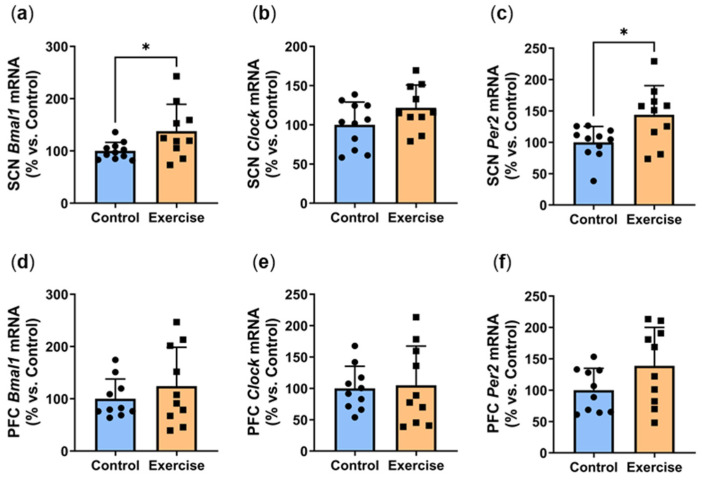
Exercise increased circadian genes at ZT7. Male *P. obesus* (n = 12/group) were split into two groups: (1) control (no running wheel) and (2) exercise (running wheel) and exposed to a short photoperiod (5 h light:19 h dark) and fed a high energy diet for 10 weeks. Suprachiasmatic nucleus (SCN) gene expression of (**a**) *Bmal1*, (**b**) *Clock* and (**c**) *Per2*. Prefrontal cortex (PFC) gene expression of (**d**) *Bmal1*, (**e**) *Clock* and (**f**) *Per2*. *Per2* mRNA levels in the (**g**) heart, (**h**) visceral adipose and (**i**) subcutaneous adipose. All gene expression data are normalised using the ^ΔΔ^Ct method to *Cyclophilin* and controls. Data presented as mean ± SD. * *p* < 0.05, ** *p* < 0.01 by unpaired t test. Statistical analyses were performed using either the unpaired *t*-test or the Mann–Whitney test.

**Table 1 ijms-25-07756-t001:** Body weights and fasting baseline glucose levels.

	Control	Exercise
Body weight (g)	217.9 ± 45.1	200.0 ± 32.9
Fasting baseline glucose (mM)	3.8 ± 0.4	4.0 ± 0.5

Data presented as mean ± SD. Statistical analysis was performed using an unpaired *t*-test.

## Data Availability

The original contributions presented in the study are included in the article. Further inquiries can be directed to the corresponding authors.

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
