# Peer review of "Exercise Reduces Glucose Intolerance, Cardiac Inflammation and Adipose Tissue Dysfunction in Psammomys obesus Exposed to Short Photoperiod and High Energy Diet"

_ijms, 2024, doi:10.3390/ijms25147756_

Round 1
Reviewer 1 Report
Comments and Suggestions for Authors
Tan et al. present a study whereby voluntary exercise (running wheel) is used to blunt the metabolic effects of a high energy diet in combination with circadian disruption. While the study is well written, comprehensive, and novel, there are significant gaps in the work that decrease enthusiasm, and especially the lack of a true healthy control. The statement is made that “the effects of exercise were tested on the backdrop of circadian disruption and a high energy diet.” Is that a fair statement to make given that your control is the diseased model and not a healthy control animal? Without that healthy control, the normalization is to the unhealthy control and thus, the impairment to circadian rhythm isn’t established and more importantly, that exercise rescues a particular alteration. Certainly, you can make the statement that “exercise upregulates circadian factors” but I’m not sure it’s appropriate to say it protects against damage when you’ve haven’t established that in this study. For instance, in the author’s prior study with the same perturbation, the level of Ccl2 in cardiac tissue did not change significantly with the HE/nocturnal paradigm, so while exercise does blunt Ccl2 it’s not really accurate to say that it protects against inflammation because it was never upregulated to start. Further, there are some questions regarding additional details in the methods. Overall, I think the work would benefit from clearer statements regarding the author’s prior work (which demonstrate expertise) and how this work provides additional information and builds on the prior publications.
Minor thoughts:
1. The last paragraph in the introduction is essentially a recap of the results and doesn’t belong in this section. Perhaps an expansion of the introduction into the markers that were assessed or the specific modality of exercise would be more appropriate.
2. Are the baseline glucose levels of 3.8-4.0 considered diabetic in this model? Is this fasting baseline glucose? If so, how long were the animals fasted? Was the OGTT conducted under fasting conditions?
3. Is the glucose intolerance in these animals based entirely on glucose clearance from the OGTT test? Other indices would also be appropriate, like adipose tissue accumulation, HBA1c’s, any assessment of a diabetic phenotype like polydipsia or polyuria.
4. What was the rational for measuring Rela alone and not including NFκB in the adipose tissue?
5. Due to the variability of the data in some of the animals, was the actual running wheel activity for each animal recorded? Further, when did the increase in activity take place in these animals? Were feeding habits or metabolism assessed? Was total adiposity, skeletal muscle, or liver size assessed?
6. Plasma insulin really needs to be assessed in these cohorts, especially since it could shed light on the overall impact of the mechanisms assessed. Further, given the decrease in adipose inflammation, plasma inflammatory variables should also be considered.
7. It is the reviewers understanding that Zeitgeber time is based on the beginning of the light phase. As such, based on the described methods, it’s challenging to understand (and thus replicate) the protocol. What is the normal day/night cycle for these animals and how was it altered relative to normal? A schematic would likely be helpful.
8. Other circadian clocks should be assessed for completeness, including CLOCK, BMAL, Per1, Cry1, and cry2.
9. What was the rational for placing the animals on the low-energy diet before the high energy? How long were they on the low-energy diet?
Author Response
Please see our responses in the attached pdf.

Reviewer 2 Report
Comments and Suggestions for Authors
General comments
This study provides interesting information, but revision is necessary before publication.
There is growing evidence that circadian disruption causes glucose intolerance, cardiac fibrosis, and adipocyte dysfunction. It is also reported that exercise intervention can improve glucose metabolism, insulin sensitivity, adipose tissue function and protect against inflammation in rodents including sand rats. As expected from accumulating knowledge, this study indicated that exercise protects against circadian disruption-induced glucose intolerance, cardiac inflammation and adipose tissue dysfunction. From a huge number of papers, the results in this study can be easily expectable. If so, what is new? What is the most important originality of this study? What is the impact of the findings in this study?
Specific comments
(1) Detailed information for the method of exercise training is necessary together with reference for the exercise method used.
1) Training period (days, weeks, months etc.)?
2) Exercise duration?
3) How many exercise training per day?, per week?, per month?
4) Intensity (low-intensity? moderate-intensity? high-intensity?) of exercise?
5) How much %VO2max ?
6) Which type of exercise? For example; low intensity exercise, high intensity exercise, resistance exercise, endurance exercise, aerobic exercise?
(2) The data of food intake must be indicated because reduction of food intake by high intense exercise might affect lipid and carbohydrate metabolisms.
(3) The data of adipose tissue weight should be indicated.
Author Response

(The authors gave the same response as above.)

Round 2
Reviewer 2 Report
Comments and Suggestions for Authors
Although the manuscript was considerably revised, the following points should be more clearly highlighted in abstract (Line 19-33) and conclusions (Line 411-416) to help readers to understand.
1. What is the most important originality in this study?
2. What is the impact of this study in this field?
